

# Development of a stem taper equation and modelling the effect of stand density on taper for Chinese fir plantations in Southern China

Aiguo Duan[1,2], Sensen Zhang[1], Xiongqing Zhang[1,2] and Jianguo Zhang[1,2]

[1] State Key Laboratory of Tree Genetics and Breeding, Key Laboratory of Tree Breeding and Cultivation of the State Forestry Administration, Research Institute of Forestry, Chinese Academy of Forestry, Beijing, PR China
[2] Collaborative Innovation Center of Sustainable Forestry in Southern China, Nanjing Forestry University, Nanjing, PR China

## ABSTRACT

Chinese fir (*Cunninghamia lanceolata*) is the most important commercial tree species in southern China. The objective of this study was to develop a variable taper equation for Chinese fir, and to quantify the effects of stand planting density on stem taper in Chinese fir. Five equations were fitted or evaluated using the diameter-height data from 293 Chinese fir trees sampled from stands with four different densities in Fenyi County, Jiangxi Province, in southern China. A total of 183 trees were randomly selected for the model development, with the remaining 110 trees used for model evaluation. The results show that the Kozak's, Sharma/Oderwald, Sharma/Zhang and modified Brink's equations are superior to the Pain/Boyer equation in terms of the fitting and validation statistics, and the modified Brink's and Sharma/Zhang equations should be recommended for use as taper equations for Chinese fir because of their high accuracy and variable exponent. The relationships between some parameters of the three selected equations and stand planting densities can be built by adopting some simple mathematical functions to examine the effects of stand planting density on tree taper. The modelling and prediction precision of the three taper equations were compared with or without incorporation of the stand density variable. The predictive accuracy of the model was improved by including the stand density variable and the mean absolute bias of the modified Brink's and Sharma/Zhang equations with a stand density variable were all below 1.0 cm in the study area. The modelling results showed that the trees have larger butt diameters and more taper when stand density was lower than at higher stand density.

Corresponding authors
Aiguo Duan, duanag@163.com
Jianguo Zhang, zhangjg@caf.ac.cn

## INTRODUCTION

Chinese fir (*Cunninghamia lanceolata*) is the most common coniferous species in southern China, occurring in both naturally regenerated stands and plantations. According to the seventh Chinese National Forest Inventory, Chinese fir plantations occupied almost 8.54 million ha and have a standing stock volume of 620.36 million m³ as a dominant tree species (*SFA, 2009*). The estimation of individual tree volume for Chinese fir is often based on existing volume tables. Volume tables are needed to accurately estimate tree volume

or merchantable timber volume at any stem diameter along the trunk in accordance with wood use in the industry through the use of compatible volume and taper equations (*Kozak, 1988*; *Riemer, Gadow & Sloboda, 1995*; *Bi, 2000*).

A stem taper equation describes a mathematical relation between tree height and the stem diameter at that height. It is thus possible to calculate the stem diameter at any arbitrary height and conversely, to calculate the tree height for any arbitrary stem diameter. Consequently, the stem volume can be calculated for any log specification and a volume equation can be developed for classified product dimensions. Numerous and various mathematical taper functions have been developed in attempts to describe tree taper. Viewed from the structures of these equations, the different taper equations can be divided into three major categories: simple mathematical equations (*Kozak, Munro & Smtth, 1969*; *Reed & Byrne, 1985*; *Pain & Boyer, 1996*; *Sharma & Oderwald, 2001*), segmented taper equations, represented by *Max & Burkhart (1976)*, *Brink & Gadow (1986)*, *Clark, Souter & Schlaegel (1991)*, *Gadow & Hui (1998)*, *Brook, Jiang & Ozcelik (2008)* and *Cao & Wang (2011)*, and variable-exponent taper equations, the latter being introduced by *Newberry & Burkhart (1986)*, *Kozak (1988)*, *Kozak (1997)*, *Kozak (2004)*, *Newnham (1992)*, *Riemer, Gadow & Sloboda (1995)*, *Bi (2000)* and *Sharma & Zhang (2004)*. Research has shown that variable-exponent taper equations performed better than the other two types of equations, and were found to be the most accurate taper equations (*Newnham, 1988*; *Kozak, 1988*; *Muhairwe, 1999*; *Sharma & Zhang, 2004*; *Rojo et al., 2005*).

Some variables related to forest management have long been recognized such as planting density, fertilization, thinning and age (*Gray, 1956*; *Bi & Turner, 1994*; *Palma, 1998*; *Sharma & Zhang, 2004*). Some tree-level or stand-level indices (e.g., crown height, ratio, and site class) have been introduced into taper equations to improve modeling performance (*Burkhart & Walton, 1985*; *Valenti & Cao, 1986*; *Newnham, 1992*; *Muhairwe, Lemay & Kozak, 1994*; *Özçelik, Diamantopoulou & Brooks, 2014*). In contrast, stand density is more easily obtained; in addition, *Sharma & Zhang (2004)* introduced stand density information into a previously developed variable taper equation for Black Spruce and found improved fit statistics and predictive accuracy. *Sharma & Parton (2009)* further modeled stand density effects on taper for Jack Pine and Black Spruce plantations using dimension analysis, and reported that the difference in bole diameter between trees at lower and higher stand densities diminished as stand density increased. *Gadow & Hui (1998)* have developed a taper equation based on the modified Brink's function for Chinese fir plantations, but no attempt has been made to quantify the stand density effect on tree taper for Chinese fir.

The objective of this study was to develop a taper equation and quantify the effect of planting density on stem taper for Chinese fir in Southern China.

## MATERIALS AND METHODS

### Data

A total of 293 trees sampled from 12 plots of even-aged Chinese fir stands were used in the present study. The trees were taken from unthinned stands that were planted in 1981 for a density-effect study of Chinese fir in Fenyi County, Jiangxi Province, of southern

**Table 1  Summary statistics for total height and dbh of Chinese fir trees used in this study.**

| Plant density (stems/ha) | Number of trees | Mean height (m) | S.D. | Mean dbh (cm) | S.D. |
|---|---|---|---|---|---|
| Fit data | | | | | |
| B: 3,333 (2 × 1.5 m) | 30 | 13.49 | 2.42 | 12.52 | 2.61 |
| C: 5,000 (2 × 1 m) | 48 | 12.85 | 2.29 | 11.03 | 2.48 |
| D: 6,667 (1 × 1.5 m) | 54 | 12.02 | 1.90 | 10.26 | 2.10 |
| E: 10,000 (1 × 1 m) | 51 | 11.83 | 2.86 | 9.62 | 2.09 |
| Validation data | | | | | |
| B: 3,333 (2 × 1.5 m) | 20 | 13.91 | 2.14 | 12.94 | 2.48 |
| C: 5,000 (2 × 1 m) | 30 | 13.25 | 2.14 | 11.47 | 2.06 |
| D: 6,667 (1 × 1.5 m) | 30 | 12.33 | 1.76 | 10.47 | 1.92 |
| E: 10,000 (1 × 1 m) | 30 | 11.93 | 1.84 | 9.30 | 1.74 |

**Notes.**
S.D. indicates standard deviation.

China (*Duan et al., 2013*). A series of stand planting densities included densities of 1,667 (A: 2 × 3 m), 3,333 (B: 2 × 1.5 m), 5,000 (C: 2 × 1 m), 6,667 (D: 1 × 1.5 m), and 10,000 (E: 1 × 1 m) stems/ha. Every planting density plot had three replications. A 2008 ice storm (*Zhou et al., 2011*) damaged or felled numerous trees in the trial plots. The sampled 293 trees were distributed in four kinds of planting densities: B, 50 trees; C, 78 trees; D, 84 trees; and E, 81 trees. Only five trees were felled in the A plots; therefore, these trees were excluded from the analysis. Two hundred and ninety three trees from different planting densities were divided into size classes based on diameter at breast height (*D*), and random selection was then applied to each of size class for data splitting, with 183 trees selected for model development, and the remaining 110 trees used for model evaluation.

The total tree height (*H*: m) and *D* (cm) were measured. Diameter outside bark (cm) was also measured at heights of 0.2, 1, 1.3, and 2 m and then at intervals of 1 m along the remainder of the stem. Table 1 summarizes the statistics related to tree characteristics.

Figure 1 shows a plot of diameter against relative height, which reflects the rate of decline in diameter with increasing height along the bole. *Chinese fir* stands mentioned in the study all are built and authorized by Research Institute of Forestry of Chinese Academy of Forestry and the data originated from our own survey. No specific permits were required for the described field data, and the field studies did not involve endangered or protected species.

## Taper equations

Five taper equations were analyzed in the present study, including two simple mathematical equations (*Pain & Boyer, 1996*; *Sharma & Oderwald, 2001*), one segmented and variable-exponent taper equation (*Riemer, Gadow & Sloboda, 1995*), and two variable-exponent taper equations (*Kozak, 2004*; *Sharma & Zhang, 2004*).

*Brink & Gadow (1986)* assumed that a tree form is composed of upper and lower parts, and developed a three-parameter equation for the whole stem taper:

$$r(h) = b_1 + (r_{1.3} - b_1) \cdot e^{b_2(1.3-h)} - \frac{b_2 i}{b_2 + b_3}(e^{b_3(h-H)} - e^{b_3(1.3-H)+b_2(1.3-h)}) \tag{1}$$

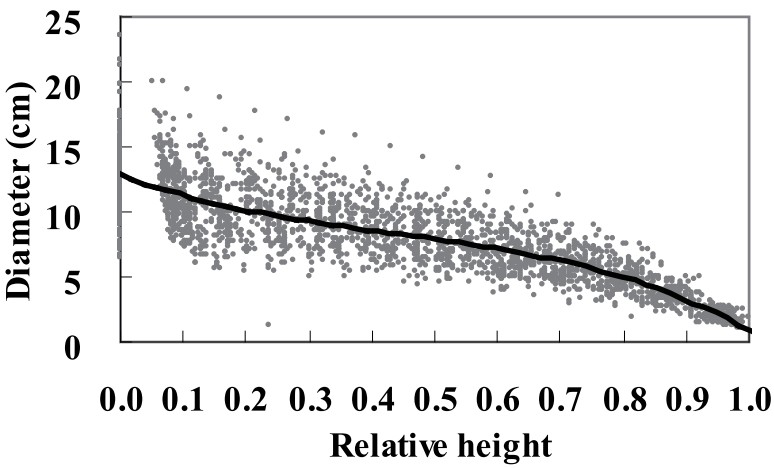

**Figure 1**  Tree diameter plotted against relative height with a cubic spline smoothing curve.

where, $r(h)$: stem radius (cm) at height $h$ (m), $h$: tree height from the ground, $H$: total height (m), $r_{1.3}$: stem radius at breast height, $b_1, b_2, b_3$ are parameters to be estimated.

Because Eq. (1) could not fulfill the condition that $r(h)$ is equal to zero when $h = H$, *Riemer, Gadow & Sloboda (1995)* proposed the modified Brink's equation:

$$r(h) = u + v \cdot e^{-b_2 h} - w \cdot e^{b_3 h} \tag{2}$$

where,

$$u = \frac{b_1}{1 - e^{b_3(1.3-H)}} + (r_{1.3} - b_1)\left(1 - \frac{1}{1 - e^{b_2(1.3-H)}}\right),$$

$$v = \frac{(r_{1.3} - b_1) \cdot e^{1.3b_2}}{1 - e^{b_2(1.3-H)}},$$

$$w = \frac{b_1 \cdot e^{-b_3 H}}{1 - e^{b_3(1.3-H)}}.$$

*Pain & Boyer (1996)* assumed that stem diameter was only determined as a function of relative height, and developed a two-parameter taper model as follows:

$$d(h) = b_1\left(1 - \left(\frac{h}{H}\right)^3\right) + b_2 \ln(h/H) \tag{3}$$

where, $d(h)$: diameter (cm) at height (m).

Based on dimensional analysis, *Sharma & Oderwald (2001)* developed a dimensionally compatible one-parameter taper equation:

$$d^2(h) = D^2\left(\frac{h}{1.3}\right)^{2-b_1}\left(\frac{H-h}{H-1.3}\right) \tag{4}$$

where, $D$: diameter at breast height.

*Sharma & Zhang (2004)* assumed that the $b_1$ in Eq. (4) could be expressed in terms of the relative height ($z$), and resulted in a variable-exponent taper equation, i.e.,

$$d^2(h) = b_1 D^2 \left(\frac{h}{1.3}\right)^{2-(b_2+b_3 z+b_4 z^2)} \left(\frac{H-h}{H-1.3}\right) \tag{5}$$

where, $z = \frac{h}{H}$, $b_4$ is parameter.

*Kozak (2004)* developed a variable-exponent taper equation as

$$d = b_1 D^{b_2} H^{b_3} \left[\frac{1-z^{1/3}}{1-p^{1/3}}\right]^{[b_4 z^4 + b_5(1/\exp(D/H)) + b_6(\frac{1-z^{1/3}}{1-p^{1/3}})^{0.1} + b_7(1/D) + b_8 H^{1-z^{1/3}} + b_9(\frac{1-z^{1/3}}{1-p^{1/3}})]} \tag{6}$$

where, $b_4$, $b_5$, $b_6$, $b_7$, $b_8$, $b_9$ and $p$ are parameters.

## Model development and evaluation with or without density variable

Equations (2)–(6) were fitted first and then compared using fit data set including 183 trees to get the suitable taper equations for Chinese fir trees. Secondly, the fit and validation data set were separately divided into the four density classes mentioned above (B, C, D and E) (Table 1), and the suitable taper equations from the first step were fitted and validated separately to each density class. To evaluate the predictive ability of the equations over the whole bole, the relative heights ($z$) were divided into ten sections for each stand planting density. Thirdly, the density variable was introduced into the selected taper equations through discussing the mathematical relationship between the resulting coefficients and the density classes or building and adding a stand density function to the exponents of the equations (*Valenti & Cao, 1986*; *Sharma & Zhang, 2004*). Lastly, the suitable taper equations with density variable were separately fitted and evaluated by the whole fit data set and the whole evaluation data set.

## Model simulation and evaluation criteria

All the equations were fitted by the NLIN procedure in the SAS statistics program (*SAS Institute, 2008*). Multicollinearity is defined as a high degree of correlation among several independent variables. The existence of multicollinearity is not a violation of the assumptions underlying the use of regression, and therefore does not seriously affect the parameter estimates and the predictive ability of the equation (*Myers, 1990*; *Kozak, 1997*). Two general methods have been suggested to deal with continuous and multilevel longitudinal data. The first is to incorporate random subject effects (*Gregoire, Schabenberger & Barrett, 1995*), and the other is to model the correlation structure directly. In the present study the first method was adopted to test the simulation properties of the taper equations in the presence of autocorrelation. Figure 2 describes the error structure of Eq. (5) with or without random subject effects incorporated. It was found that the simulation result hadn't been obviously altered while considering autocorrelation. Additionally, some studies had found that the Eqs. (2) and (6) showed very low multicollinearity (*Rojo et al., 2005*; *Kozak, 1997*). Therefore, the correlated error structure in the data was not considered in the SAS MODEL procedure.

The model adjusted coefficient of determinations ($R^2_{\text{adj.}}$), mean difference (bias: M.D.), mean absolute difference (M.A.D.) and standard error of estimate (S.E.E.) were examined

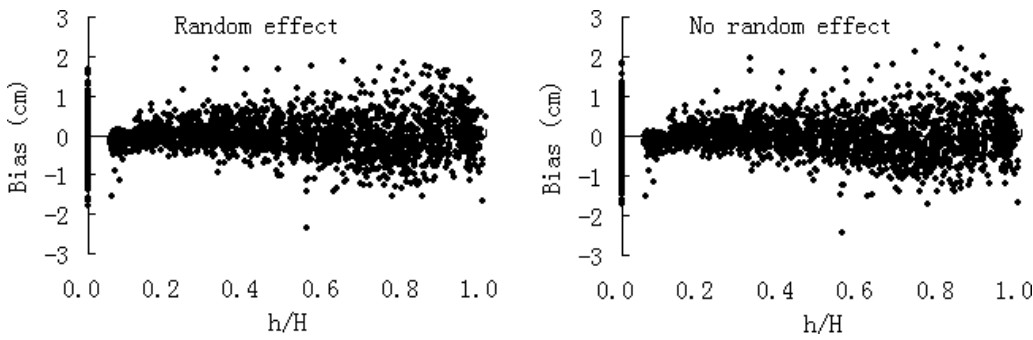

**Figure 2** The error structure of Eq. (5) with or without random subject effects incorporated.

while comparing modeling accuracy of the equations. These statistical indices can be calculated using Eqs. (7)–(10):

$$R^2_{adj.} = 1 - \frac{\frac{1}{n-k-1}\sum_{k=1}^{n}(obs_k - est_k)^2}{\frac{1}{n-1}\sum_{k=1}^{n}(obs_k - \overline{obs_k})^2} \tag{7}$$

$$M.D. = \sum_{k=1}^{n}\frac{obs_k - est_k}{n} \tag{8}$$

$$M.A.D. = \sum_{k=1}^{n}\frac{|obs_k - est_k|}{n} \tag{9}$$

$$S.E.E. = \sqrt{\frac{\sum_{k=1}^{n}(obs_k - est_k)^2}{n - m}} \tag{10}$$

where $obs_k$ and $est_k$ are the observed and predicted diameter along the bole for the $k$th height point, respectively, $n$ is the number of height points along the bole, and $m$ is the number of equation parameters.

## RESULTS AND DISCUSSION

### Without stand density

Table 2 presents the fit statistics and parameters of Eqs. (2)–(6) using the fit data. Based on $R^2_{adj.}$ and S.E.E., the Kozak equation, Sharma/Zhang, Sharma/Oderwald and modified Brink's equations have higher precision than the Pain/Boyer equation. The S.E.E. of Kozak, Sharma/Zhang, Sharma/Oderwald and modified Brink's equations were 0.5194, 0.5224, 0.5335 and 0.6629, respectively. The results proved that the variable-exponent taper equations (Eqs. (2), (5) and (6)) all had the higher modelling precision than the simple mathematical taper equation (Eq. (3)). It is worth noting that the simple mathematical equation (Eq. (4)) also had high modelling precision for Chinese fir tree's stem taper.

The accuracy of diameter predictions by these five taper equations was evaluated along the bole of Chinese fir trees using the validation data sets (Fig. 3). Diameter prediction bias of Chinese fir trees for Eqs. (2) and (4)–(6) was smaller than for Eq. (3). Obviously, the predicted diameter corresponding to the section closest to the ground was generally underestimated for all of the five equations. When compared with the

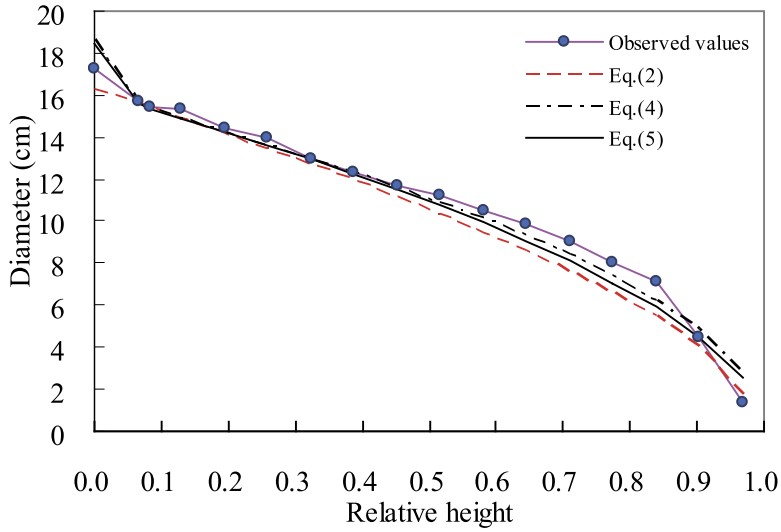

**Figure 3** The observed stem taper and tree profiles generated using **Eqs. (2), (4)** and **(5)** for a Chinese fir tree with the diameter at breast height (15.4 cm) and total height (15.5 m).

**Table 2** Values of the estimated parameters and fit statistics for **Eqs. (2)–(6)** fit to taper data for all of the 183 trees (ns means not significance at the 95% level).

| Model | $b_1$ | $b_2$ | $b_3$ | $b_4$ | $b_5$ | $b_6$ | $b_7$ | $b_8$ | $b_9$ | $p$ | $R^2_{adj.}$ | S.E.E. |
|---|---|---|---|---|---|---|---|---|---|---|---|---|
| Eq. (2) | 1.6318 | −0.0294 | 0.9428 | | | | | | | | 0.9649 | 0.6629 |
| Eq. (3) | 9.5276 | −0.3199 | | | | | | | | | 0.6952 | 1.9540 |
| Eq. (4) | 2.0307 | | | | | | | | | | 0.9772 | 0.5335 |
| Eq. (5) | 0.9964 | 2.0294 | −0.0302 | 0.1051 | | | | | | | 0.9782 | 0.5224 |
| Eq. (6) | 1.2327 | 0.9968 | ns | 0.2408 | ns | 0.5006 | −0.4297 | ns | ns | ≈0 | 0.9785 | 0.5194 |

Sharma/Oderwald and Sharma/Zhang equations, we found the modified Brink's equation only had relatively low prediction precision at the butt diameter. Except for the butt part, the accuracy of diameter predictions of the modified Brink, Sharma/Oderwald and Sharma/Zhang equations showed a trend of decrease with the increase of relative height, and the results from the three equations were very similar, with an average size of error in diameter predictions below 0.2 cm. For the Kozak equation, most of the predictions were underestimated, especially corresponding to the lower stem, which led to this equation had a relatively large error near to 1.0 cm. The reason that the Kozak equation had low accuracy at the stage of prediction might be due to its some unstable parameters. *Rojo et al. (2005)* had found that parameter $b_8$ of the Kozak equation had not significance at the 95% test level for maritime pine.

Based on the fitting and validation statistics, the Sharma Oderwald, Sharma/Zhang and modified Brink's equations are suggested for use as taper equations for Chinese fir trees. Figure 4 showed a simulation result of the three equations for a Chinese fir tree's stem taper. Besides the structural difference, the fact that the five taper equations use different sets of predictor variables may be an important reason for the differences in simulation accuracy. Equations (2) and (4) use $D$ and $H$, together with $h$. Furthermore, Eq. (3) uses

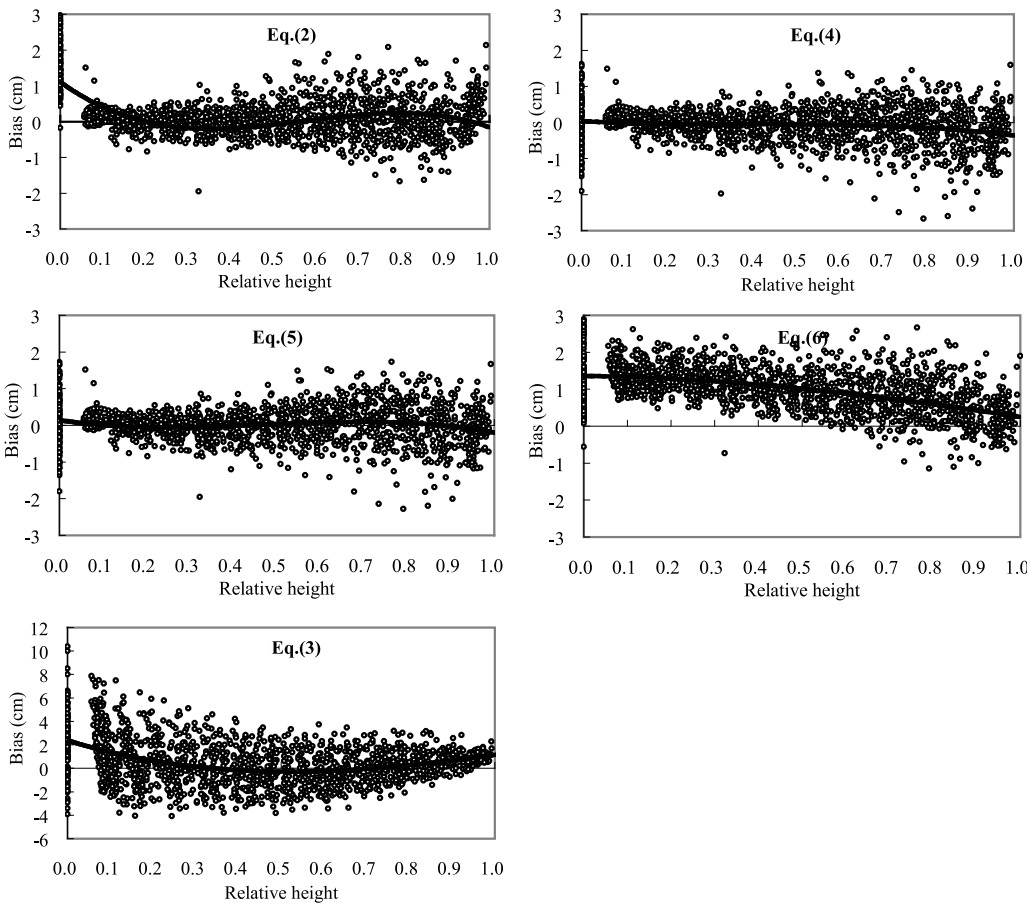

**Figure 4  Bias of taper prediction at relative height for Eqs. (2)–(6) using validation data sets.**

$H$ and $h$, but not $D$, Eq. (6) uses $D$, $H$ and the $z$, while Eq. (5) uses $D$, $H$, $h$ and also the $z$. Since $D$ is the most important factor for measuring tree size, ignoring this predictor variable may lead to the poorest performance of Eq. (3), the Pain/Boyer taper equation. In contrast, Eq. (5), the Sharma/Zhang taper equation, was shown to have the highest prediction accuracy because it included all important predictor variables (Fig. 3).

## Estimations that consider stand density

Table 3 lists the estimated parameter values and the corresponding fit statistics of Eqs. (2) and (4)–(5) for the density-grouped subsets of data. The estimates of most parameters of Eqs. (2) and (4)–(5) were significantly different between the four planting densities ($p < 0.0001$). In the case of Eq. (5), however, only the estimates for $b_2$ were significantly different between any two of the four planting densities ($p < 0.0001$). The estimates for $b_4$ were significantly different between the four planting densities ($p < 0.0001$), and the estimates for $b_1$ were not significantly different between any two of the four planting densities ($p > 0.3784$). The estimates for $b_3$ were found having no significance at the 95% level. These comparisons were made based on the confidence limits of the parameters obtained by nonlinear regressions.

**Table 3  Parameter estimates (standard errors in parentheses) and fit statistics for Eqs. (2) and (4)–(5) while fitting the density-grouped subsets data using nonlinear regression (ns means not significant at the 95% level).**

| | Parameter | Planting density (stems/ha) | | | |
|---|---|---|---|---|---|
| | | 3,333 | 5,000 | 6,667 | 10,000 |
| Eq. (2) | $b_1$ | 2.1568(0.1928) | 1.5143(0.1079) | 1.6975(0.1371) | 1.4346(0.0984) |
| | $b_2$ | 0.0011(0.0094) | −0.0436(0.0069) | −0.0243(0.0097) | −0.0392(0.0078) |
| | $b_3$ | 0.6400(0.0781) | 1.2003(0.1722) | 0.8462(0.0963) | 1.1461(0.1409) |
| | $R^2_{adj.}$ | 0.9761 | 0.9541 | 0.9653 | 0.9635 |
| | S.E.E. | 0.6455 | 0.8007 | 0.6236 | 0.6009 |
| Eq. (4) | $b_1$ | 2.0324(0.0015) | 2.0319(0.0013) | 2.0303(0.0010) | 2.0278(0.0011) |
| | $R^2_{adj.}$ | 0.9815 | 0.9713 | 0.9810 | 0.9765 |
| | S.E.E. | 0.6161 | 0.6198 | 0.4565 | 0.4768 |
| Eq. (5) | $b_1$ | 0.9976(0.0077) | 0.9893(0.0085) | 0.9998(0.0064) | 0.9973(0.0074) |
| | $b_2$ | 2.0280(0.0015) | 2.0326(0.0016) | 2.0293(0.0012) | 2.0273(0.0014) |
| | $b_3$ | ns | ns | −0.0712(0.0347) | ns |
| | $b_4$ | 0.1625(0.0509) | 0.0557(0.0551) | 0.1561(0.0443) | 0.0741(0.0513) |
| | $R^2_{adj.}$ | 0.9848 | 0.9712 | 0.9817 | 0.9769 |
| | S.E.E. | 0.4978 | 0.6180 | 0.4426 | 0.4724 |

To examine the effect of stand planting density on tree taper, the correlations between some parameters of Eqs. (2) and (4) and stand planting density were analyzed (Fig. 5). The relationships of parameter $b_1$ of Eq. (2) and parameter $b_1$ of Eq. (4) to stand planting density were well approximated by both an exponential and a linear function. The coefficients of determination ($R^2$) between parameter $b_1$ of Eqs. (2) and (4) and stand planting density were 0.68 and 0.98, respectively, and the test result of correlation coefficient showed that parameter $b_1$ of Eqs. (2) and (4) were significantly related to stand planting density ($p < 0.1$ and $p < 0.01$, respectively). The other parameters all lacked obvious monotonic correlations to stand density. The results showed that stand planting density had an obvious effect on some parameters used in the taper equations. So, the resultant parameter prediction equation for predicting $b_1$ of Eqs. (2) and (4) can be given by Eqs. (11) and (12).

$$b_1 = i_1 \cdot \mathrm{spd}^{i_2} \tag{11}$$

$$b_1 = k_1/\mathrm{spd} + k_2 \tag{12}$$

where spd refers to stand planting density, and $i_1$, $i_2$, $k_1$ and $k_2$ are parameters to be estimated. Equations (11) and (12) can be substituted into Eqs. (2) and (4), and the Eqs. (13) and (14), including a stand planting density term, are then deduced.

$$r(h) = \frac{i_1 \cdot \mathrm{spd}^{i_2}}{1 - e^{b_3(1.3-H)}} + (r_{1.3} - i_1 \cdot \mathrm{spd}^{i_2})\left(1 - \frac{1}{1 - e^{b_2(1.3-H)}}\right)$$
$$+ \frac{(r_{1.3} - i_1 \cdot \mathrm{spd}^{i_2}) \cdot e^{1.3b_2}}{1 - e^{b_2(1.3-H)}} \cdot e^{-b_2 h} - \frac{i_1 \cdot \mathrm{spd}^{i_2} \cdot e^{-b_3 H}}{1 - e^{b_3(1.3-H)}} \cdot e^{b_3 h} \tag{13}$$

$$d^2(h) = D^2 \left(\frac{h}{1.3}\right)^{2-(k_1/\mathrm{spd}+k_2)} \left(\frac{H-h}{H-1.3}\right). \tag{14}$$
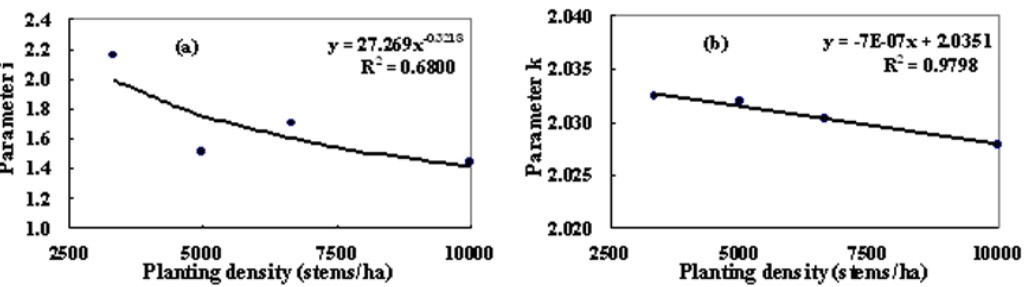

**Figure 5  The correlativities between some parameters of Eqs. (2) and (4) and stand planting densities.**

**Table 4  Parameter estimates (standard errors in parentheses) and fit statistics for Eqs. (13)–(15) using fit data sets.**

|  | Parameter | Estimates of parameters | S.E.E. | $R^2_{adj.}$ |
|---|---|---|---|---|
| Eq. (13) | $i_1$ | 2.1504 (0.5675) | 0.6627 | 0.9650 |
|  | $i_2$ | −0.0306 (0.0292) |  |  |
|  | $b_2$ | −0.0282 (0.0040) |  |  |
|  | $b_3$ | 0.9331 (0.0575) |  |  |
| Eq. (14) | $k_1$ | 21.7021 (8.5436) | 0.5327 | 0.9773 |
|  | $k_2$ | 2.0266 (0.0017) |  |  |
| Eq. (15) | $b_1$ | 0.9964 (0.0039) | 0.5219 | 0.9782 |
|  | $b_2$ | 2.0257 (0.0017) |  |  |
|  | $b_3$ | −0.0310 (0.0204) |  |  |
|  | $b_4$ | 0.1058 (0.0259) |  |  |
|  | $b_5$ | 19.6487 (8.3747) |  |  |

The test results of correlation coefficients showed that the parameters of Eq. (6) were found not having significant relevance to stand planting density ($p > 0.1$). *Sharma & Zhang (2004)* modified Eq. (6) to accommodate stand density effect by adding a stand density function to the exponent of Eq. (6), i.e.,:

$$d^2(h) = b_1 D^2 \left(\frac{h}{1.3}\right)^{2-(b_2+b_3z+b_4z^2+b_5/\mathrm{spd})} \left(\frac{H-h}{H-1.3}\right). \tag{15}$$

To examine the effect of stand planting density on tree taper in a Chinese fir plantation, Eqs. (13)–(15) were fitted to the entire fitting data set composed of different stand densities. Two fit statistics ($R^2_{adj.}$ and S.E.E.) both indicated that diameter outside bark taper equations (13)–(15) all performed well for Chinese fir trees. The S.E.E. of Eqs. (13)–(15) ranged from 0.7225 to 0.8139 cm. The accuracy in modeling tree taper was improved for all of the selected three taper equations by incorporating a stand density term in the equations (Table 4).

Equations (13)–(15) were further evaluated using the validation data sets. The bias distribution ranges of Eqs. (13)–(15) with the density variable were −0.0556 to 0.7662, −0.2779 to 0.0233 and −0.0983 to 0.1513, respectively (Table 5). Clearly, adding the stand density variable improved the evaluation efficacy for Chinese fir trees (Table 6).

**Table 5** Bias (cm) and absolute bias (cm) in predicting diameters along the bole of Chinese fir trees for Eqs. (13)–(15) (with stand density variable) using validation data sets.

| Relative height | Number | Bias (cm) | | | Absolute bias (cm) | | |
|---|---|---|---|---|---|---|---|
| | | Eq. (13) | Eq. (14) | Eq. (15) | Eq. (13) | Eq. (14) | Eq. (15) |
| $0.0 \leq h/H \leq 0.1$ | 279 | 0.7662 | 0.0233 | 0.0829 | 0.7835 | 0.3214 | 0.3250 |
| $0.1 < h/H \leq 0.2$ | 205 | −0.0226 | −0.0185 | −0.0053 | 0.1164 | 0.1170 | 0.1228 |
| $0.2 < h/H \leq 0.3$ | 157 | −0.0215 | −0.0453 | −0.0379 | 0.1962 | 0.2054 | 0.2048 |
| $0.3 < h/H \leq 0.4$ | 151 | −0.0297 | −0.0984 | −0.0757 | 0.2683 | 0.2780 | 0.2745 |
| $0.4 < h/H \leq 0.5$ | 151 | 0.0414 | −0.0795 | −0.0237 | 0.3138 | 0.3089 | 0.3043 |
| $0.5 < h/H \leq 0.6$ | 153 | 0.0976 | −0.0598 | 0.0454 | 0.3910 | 0.3662 | 0.3656 |
| $0.6 < h/H \leq 0.7$ | 154 | 0.1041 | −0.0419 | 0.1170 | 0.4514 | 0.4164 | 0.4360 |
| $0.7 < h/H \leq 0.8$ | 160 | 0.0624 | −0.0564 | 0.1513 | 0.4992 | 0.4817 | 0.5161 |
| $0.8 < h/H \leq 0.9$ | 152 | −0.0556 | −0.2528 | −0.0177 | 0.4773 | 0.5498 | 0.5106 |
| $0.9 < h/H \leq 1.0$ | 139 | 0.2573 | −0.2779 | −0.0983 | 0.4725 | 0.5370 | 0.4635 |

**Table 6** Values of the statistics in the validation step for Eqs. (2) and (4)–(5) with and without the density variable in the model.

| Statistics | Equations with stand density variable | | | Equations without stand density variable | | |
|---|---|---|---|---|---|---|
| | Eq. (13) | Eq. (14) | Eq. (15) | Eq. (2) | Eq. (4) | Eq. (5) |
| M.D. | 0.1601 | −0.0752 | 0.0223 | 0.1612 | −0.0758 | 0.0226 |
| M.A.D. | 0.4155 | 0.3452 | 0.3416 | 0.4157 | 0.3453 | 0.3418 |
| S.E.E. | 0.6543 | 0.5007 | 0.4835 | 0.6548 | 0.5011 | 0.4838 |

The modified Brink's equation mostly had larger and more positive bias at the butt and tip of tree stems than at mid-stem. However, the Sharma/Oderwald and Sharma/Zhang equations had larger bias at lower stem parts, and the Sharma/Oderwald equation had an almost negative bias along the boles excepting at the butt. The results show that the diameters at two ends of the stems of Chinese fir trees will be underestimated when using the modified Brink's equation, and were mostly overestimated by the Sharma/Oderwald equation (Table 5).

The maximum mean absolute bias of the modified Brink's, Sharma/Oderwald and Sharma/Zhang equations with the stand density variable were 0.7662, 0.5498, and 0.5161 cm. Note that the modified Brink's equation had relatively larger bias than the Sharma/Zhang and Sharma/Oderwald equations only because of the great bias at the butt of tree stems. Considering the variable-exponent taper equation's theoretical property, the modified Brink's and Sharma/Zhang equations were the most appropriate equations for describing tree taper of Chinese fir trees.

The effect of stand planting density was analyzed visually by generating tree profiles using Eq. (12) for $D = 11.0$ cm and $H = 15.0$ m at four different stand densities (1,000, 2,000, 3,000, and 4,000 trees/ha) (Fig. 6). The results show that the trees have larger butt diameters and more taper when stand density was lower than at higher stand density.

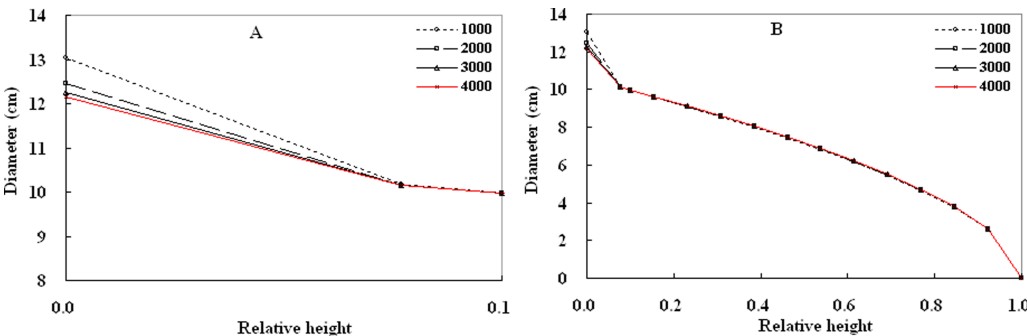

**Figure 6** Tree profiles generated from Eq. (15) using *D* = 11.0 cm and *H* = 12.0 m at different densities (1,000, 2,000, 3,000 and 4,000 trees/ha) for Chinese fir. (A) showing the difference of tree profiles below 0.1 at four different densities.

Additionally, the difference in bole diameter between trees at both lower and higher stand densities decreases as stand density increases. This phenomenon confirms the findings of *Sharma & Zhang (2004)* and *Sharma & Parton (2009)*. However, except for the butt diameters, prediction diameters in the middle section only have weak differences among the different stand densities, which was different from a study of black spruce (*Sharma & Zhang, 2004*). The reason may lie in the difference between the two tree species. *Sharma & Zhang (2004)* found that density affected the taper of jack pine more than that of black spruce. Additionally, because stand site can directly affect the diameter and high growth of trees, site may also influence tree volume (*Muhairwe, Lemay & Kozak, 1994*). However, Eq. (15), as a variable taper equation that includes a stand density variable, can well predict diameters along the boles and to a certain extent, express the effect of stand density on stem tapers of Chinese fir trees.

## CONCLUSIONS

Variable taper equations were developed for Chinese fir, the most important commercial tree species in southern China. The Sharma/Oderwald, Sharma/Zhang, and modified Brink's equations are superior to the Pain/Boyer equation in terms of the fitting and validation statistics. The modified Brink's equation only had lower prediction precision than the Sharma/Oderwald and Sharma/Zhang equations at the butt diameter. If the final choice must be made, the modified Brink's equation and Sharma/Zhang equation are recommended for use as a taper equation for Chinese fir.

Correlation analysis results showed that stand planting density had an obvious effect on some parameters of taper equations. Therefore, the relationships between some parameters of the three selected equations and stand planting densities can be built by adopting some simple mathematical functions to examine the effect of stand planting density on tree taper.

The prediction precision of the three taper equations was compared with or without incorporation of the stand density variable. The M.D., A.M.D., and S.E.E. using for estimating diameters along the stems for the validation data sets showed that adding the stand density variable improved the evaluation efficacy of the taper equations for Chinese fir trees. The maximum mean absolute bias of the modified Brink's and Sharma/Zhang

equations with a stand density variable were all below 1.0 cm in the study area. The modelling difference of tree profiles among different stand densities mainly appeared below the 10% of total high.

## ACKNOWLEDGEMENTS

Thanks go to Mr. Quang V. Cao in the Louisiana State University for his help with revising and suggestions. Special thanks go to two careful and warmhearted reviewers.

### Funding

The Scientific and Technological Task in China (Grant No. 2015BAD09B01, 2012BAD01B01), the National Natural Science Foundation of China (Grant No. 31370629) and the Collaborative Innovation Center of Sustainable Forestry in Southern China of Nanjing Forestry University supported this study. The funders had no role in study design, data collection and analysis, decision to publish, or preparation of the manuscript.

### Grant Disclosures

The following grant information was disclosed by the authors:
Scientific and Technological Task in China: 2015BAD09B01, 2012BAD01B01.
National Natural Science Foundation of China: 31370629.
Collaborative Innovation Center of Sustaintable Forestry in Southern China of Nanjing Forestry University.

### Competing Interests

The authors declare there are no competing interests.

### Author Contributions

- Aiguo Duan conceived and designed the experiments, performed the experiments, analyzed the data, contributed reagents/materials/analysis tools, wrote the paper, prepared figures and/or tables, reviewed drafts of the paper.
- Sensen Zhang performed the experiments, contributed reagents/materials/analysis tools.
- Xiongqing Zhang contributed reagents/materials/analysis tools.
- Jianguo Zhang conceived and designed the experiments, reviewed drafts of the paper.

### Data Availability

  The raw data has been supplied as Data S1.

### Supplemental Information

Supplemental information for this article can be found online at http://dx.doi.org/10.7717/peerj.1929#supplemental-information.

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
