# Peer review of "Development of a stem taper equation and modelling the effect of stand density on taper for Chinese fir plantations in Southern China"

_PeerJ, doi:10.7717/peerj.1929_

## Round 0.1 · original submission · Major Revisions

Both referees see value in your paper but they also raise major issues - first of all with the statistical analyses. You need to convince the reader that your estimates of parameters' uncertainty are robust, and that the test statistics you use are grounded in statistical theory.

Reviewer 1 ·

Basic reporting

No comments

Experimental design

In my opinion, the main concern of the paper is the lack of proper statistical procedures for fitting the taper functions. The authors should consider the problems associated with stem taper function analysis that violates the fundamental least squares assumptions, of which multicollinearity and autocorrelation are the most important
Authors also do not use an appropriate statistical method to assess if parameter estimates of taper equations are significantly different among density classes

Validity of the findings

Data are appropiate

Additional comments

Manuscript ID: peerj-review-5240

General Comments

The purpose of this study is to develop a taper equation and quantify the effect of stand density on stem taper for Chinese fir in Southern China. The originality in the paper is a new stem taper equation for Chinese fir plantations in Southern China. There is nothing new concerning methods.
It is well known that single taper functions are worse than segmented and variable-form taper functions. So, why do the authors evaluate two single taper functions? In fact in the abstract the authors state as an aim of the work the development of a variable taper equation. It would be better to evaluate more segmented and variable-form taper functions. For example, the variable-exponent model proposed by Kozak (2004) or the segmented compatible model developed by Fang et al. (2000) have successful applied to different species (e.g. Corral-Rivas et al., 2007; Diéguez-Aranda et al., 2006).
In my opinion, the main concern of the paper is the lack of proper statistical procedures for fitting the taper functions. The authors should consider the problems associated with stem taper function analysis that violates the fundamental least squares assumptions, of which multicollinearity and autocorrelation are the most important (Kozak, 1997). Multicollinearity refers to the existence of high intercorrelations among the independent variables in multiple linear or nonlinear regression analysis, because some of the variables represent of measure similar phenomena. Autocorrelation is due to the fact that the database used for model fitting contains multiple observations for each tree (i.e., hierarchical data), so it is reasonable to expect that the observations within each tree are spatially correlated, which violates the assumption of independent error terms. Although the least squares estimates of regression coefficients remain unbiased and consistent under the presence of multicollinearity and autocorrelation, they are no longer efficient (Myers, 1990, p. 288). These problems may seriously affect the standard errors of the coefficients, invalidating statistical tests using t or F distributions and confidence intervals (Neter et al., 1990, p. 300). Thus, appropriate statistical procedures should be used in model fitting to avoid problems of autocorrelated errors (see, for example, Zimmerman and Núñez Antón, 2001), and models with low multicollinearity should be selected whenever possible (Kozak, 1997).
Authors also do not use an appropriate statistical method to assess if parameter estimates of taper equations are significantly different among density classes. For this subject you can use the nonlinear extra sum of squares method (Bates and Watts, 1988, pp. 103-104). This method requires the fitting of full and reduced models and has frequently been applied to assess if separate models are necessary for different species or different levels of a classification variable (e.g. Huang et al., 2000). The reduced model corresponds to the same set of global parameters for all density classes whereas the full model corresponds to different sets of parameters for each density class and it is obtained by expanding each parameter including an associated parameter and a dummy variable to differentiate among density classes.
Therefore, taking into account the above considerations, the paper could be accepted for publication after I consider major revision


References cited in the Reviewer’s Comment / Report
Bates D.M. and Watts D.G., 1988. Nonlinear regression analysis and its applications, Wiley, New York.
Corral-Rivas J.J., Diéguez-Aranda U., Corral S., and Castedo F. (2007).A merchantable volumen system for major pine species in El Salto,Durango (Mexico). For. Ecol. Manage. 238: 118–129.
Fang, Z., Borders, E., and Bailey, L. 2000. Compatible volume-taper models for loblolly and slash pine based on a system with segmented-stem form factors. For. Sci. 46(1), 1-12.
Diéguez-Aranda U., Castedo F., Álvarez J.G., and Rojo A., 2006. Compatible taper function for Scots pine plantations in northwestern Spain. Can. J. For. Res. 36: 1190–1205.
Huang, S., Price, D., Morgan, D. and Peck, K. (2000) Kozak’s variable-exponent taper equation regionalized for white spruce in Alberta. West. J. Appl. For. 15(2): 75–85.
Kozak, A., 1997. Effects of multicollinearity and autocorrelation on the variable-exponent taper functions. Can. J. For. Res. 27, 619–629.
Kozak A. (2004). My last words on taper equations, For. Chron. 80: 507–515.
Myers, R.H., 1990. Classical and Modern Regression with Applications, second ed. Duxbury Press, Belmont, California.
Neter, J., Wasserman, W., Kutner, M.H., 1990. Applied Linear Statistical Models: Regression Analysis of Variance and Experimental Designs, third ed. Irwin, Boston.
Zimmerman, D.L., Nuñez-Antón, V., 2001. Parametric modelling of growth curve data: an overview (with discussion). Test 10, 1–73.


Specific Comments


Title
Consider modifying the title of the paper “Development of a stem taper equation and modelling the effect of stand density on taper for Chines fir plantation in Southern China”

Results and discussion
(1) Lines 190-195. How the comparison among parameters estimated was made? How the authors state that parameter estimates are significantly different?
(2) Line 281: Replace “many” with “may”
(3) Table 3: Eq. (8) have clearly not significant parameters in some planting densities (e.g. 2 and 3)
(4) Line 470: Replace “Eqs. (11-13)” with “Eqs. (15-17)”
(5) Table 4: Replace “Eq. (11), Eq.(12) and Eq. (13)” with “Eq. (15), Eq.(16) and Eq. (17)”, respectively.
(6) Line 635: Replace “Eq. (13)” with “Eq. (17)”
(7)


Results
(1) Page 8/line 46-47: Replace “and average distance from forest to women’s residence was longer (54 km, 39 km)” with “and average distance from forest to women’s residence was longer (54 km) than the corresponding for men (39 km)”


References
(1) Ramaza et al. (2014) citation was not included in the list of references.
(2) The next references are not cited in the text and must be dropped from the reference list:
 Honer, T.G., Standard volume tables and merchantable conversion factors for the commercial tree species of central and eastern Canada. Information report FMR-X-5. Forest Management Research and Services Institute, Ottawa, Ont., Canada, 1967, 78p.
 Kozak, A., My last words on taper equations. Forestry Chronicles, 2004, 80(4): 507-515.
 Li, R.X., Weiskittel, A., Dick, A.R., Kershaw, J., and Seymour, R.S., Regional stem taper equations for eleven conifer species in the acadian region of north America: development and assessment. North. J. Appl. For., 2012, 29(1):5-14.
 Özçelik, R, Diamantopoulou, M.J., and Brooks, J.R., The use of tree crown variables in over-bark diameter and volume prediction models. iForest, 2014, 7: 132-139

Reviewer 2 ·

Basic reporting

see blow

Experimental design

see blow

Validity of the findings

see blow

Additional comments

Tree taper equation research is a very interesting topic and plenty of research has been documented in the literature. This manuscript attempted to compare and select an appropriate taper equation form for Chinese fir plantation through a comparative study using 4 different existing taper equations, and incorporate plantation density into the selected taper equation. This research subject and contents are meaningful and potentially useful in forest management planning. However, a major revision is needed to be re-considered for publication because of the following issues:

•The ultimate goal of studying tree taper equations is to search the best equation form that can characterize the diameter-height relationship of individual trees, for the purpose of improving accuracy in volume estimation (both total and merchantable). In other words, to select the best equation that can improve volume estimation. The authors need to clarify the research applications of this study: for academic or practical forestry. Current version of manuscript seems targeted at academic exercise because it stopped at comparing the statics from data fitting, and stopped short on how the different statics could translate into the improvement in accuracy of volume estimation.
•For the academic research purpose, the authors need to show their results are fully supportive to the main conclusion. However, this was not true in current version such as the figure 4 showed equation 13 can predict almost identical the diameters along different relative heights under different stand densities, except the butt portion of trees. This figure does not support the main purpose of this study that “modeling the effects of stand density on taper”.
•Last, but not the least is the concern over English writing style. This manuscript seemed not following the usual manuscript style in English writing. Examples include:
•Confusing presentations: Tree samples in the Abstract needs to be made clear: not all the 293 trees were used in fitting relationship as the authors described in L28-32 “Four equations were fitted to diameter-height data from 293 Chinese fir trees…183 trees were … for the model development…”;
•L40-41: “maximum mean absolute bias”, which was never defined in the manuscript;
•L54-55: unclear about “any specified upper stem diameter”, did the authors mean “diameter at top or merchantable height” defined by any specific utilization standard? If yes, the authors should also mention “stump height” which may have even larger impact on merchantable volume estimation than the diameter at top/merchantable height;
•L56-59: these two sentences need rewording;
•L67-68: suggest changing to “research has showed that variable-exponent taper equations performed better than that of other two types of equations”;
•L70: suggest changing to “some variables related to forest management have long been recognized such as …”;
•L81-82: delete the sentence of “however … Chinese fir”;
•L85-88: Research objective(s) need to be made clear: one or more?
•L89: suggest changing to “Materials and methods”;
•L108-110: Figure 1 can be removed without influencing the rest of manuscript;
•Planting densities and their labels need to be consistent across the manuscript: in L94, the authors defined A as density of 1667, but this density was not appeared in results/tables; the authors defined D as density of 6667 in L95, and stated this density was not used in analysis because only 5 trees were fell down (L98-99); nevertheless, this density was used in the analytical results and tables;
•L118-130: equations 1-3 can be removed since the description of developmental history of Riemer et al. (1995) equation seems contributing insignificantly to this study;
•L150-153: the authors need to clarify the purpose/benefit of rearranging equation 7 as equation 8 for being used in current study: easy in parameter estimation, or standardization?
•L154: this section “Inclusion of stand density” needs to be completely rewritten – current version is unclear and confusing. The authors stated steps how they did the analysis, without describing the purpose for each step. For example,
•L155 actually stated how the authors handled the situation of without including stand density, furthermore, the authors also need to specify only 183 tree were used in this step and the remaining trees were used in validation;
•L156 needs to specify the two categories of each of the four planting densities; therefore, total 8 subsets of data were formed;
•L157 unclear how the four forms of equations were used for both “fitted and validated” purpose;
•L158 “ten sections”, are they referred to relative heights?
•L160-161: unclear about the effect of using other data sets on the current results of detecting trend;
•L161-162: unclear why “resulted in many new taper equations”;
•L162-164: unclear how the authors did this;
•L170 “examined” should be “compared”;
•L171: “These parameters” – these are statics, not parameters;
•L188: Table 2: “for all trees” – unclear this was referred to all 293 trees or 183 trees. Furthermore, the average R2adj and S.E.E. appeared much better than the ones presented in Table 3 (stand density effect are included). If this is the case, the main conclusion from this manuscript needs to be re-considered;
•More items can be added onto this list, but I choose stopping here. In summary, due to writing styles in different languages, current version needs a large effort in language polishing.

---

## Round 0.2 · Major Revisions

You have not addressed the main concern that was raised by referee #1 and myself:

I wrote "Both referees see value in your paper but they also raise major issues - first of all with the statistical analyses. You need to convince the reader that your estimates of parameters' uncertainty are robust, and that the test statistics you use are grounded in statistical theory." and Referee 1 wrote (his/her comment 3): "In my opinion, the main concern of the paper is the lack of proper statistical procedures for fitting the taper functions. The authors should consider the problems associated with stem taper function analysis that violates the fundamental least squares assumptions, of which multicollinearity and autocorrelation are the most important"

You answer is "The contents in the context are “multicollinearity is defined as a high degree of correlation among several independent variables. The existence of multicollinearity is not a violation of the assumptions underlying the use of regression, and therefor does not seriously affect the predictive ability of the equation (Myers, 1990; Kozak, 1997). So the correlated error structure in the data was not considered in the SAS MODEL procedure.” However, we think this explain still can not solve the valuable suggestion. But we sincerely hope this revised can get the reviewer’s comprehension. We will furtherly study and make efforts to improve the quality in the future studies."

The first part of your answer is somewhat wrong - multicollinearity is a serious concern (see referee's comment), because it leads among, other things, to unstable parameter estimates and predictions. Second, you chose to ignore the problem of having model residuals that could be strongly dependent (ie autocorrelated), and as the referee wrote (and this IS an important issue), this will affect the uncertainty of your parameter estimates, which model is selected and therefore your predictions. There exist many ways to include this in your models (eg in the free software R in the nlme function). You cannot just answer that will do it in another paper.

---

## Round 0.3 · Minor Revisions

The current revised paper is a significant improvement over the previous version, and there are only a few remaining minor changes that need to be made:

l. 78-79: Research has shown
l. 255-56: "For the Pain/BoyerKozak equation, the most predictions were underestimated". What do you mean by "most predictions"? All predictons?
l. 270: replace proven by shown
l. 280-281, 290, 305: Would be best to provide exact P-values rather than </>0.05
l. 289: give R2 with only two digits, and for other criteria 3 digits are sufficient
Table 3: not significant at the...
Figure 4: Correlations between ... The line for a) is not linear whereas the model shown is linear
Figure 5: consider a change a scale to show differences at height < 0.1 more explicitly

---

## Round 0.4 · accepted · Accept

Changes were made to take into account my last comments, the paper is now ready for publication